Disturbed neurovascular coupling in hemodialysis patients

Jin Mei 1
Wang Liyan 2
Wang Hao 1
Han Xue 2
Diao Zongli 2
Guo Wang 2
Yang Zhenghan 1
Ding Heyu 1
Wang Zheng 1
Zhang Peng 1
http://orcid.org/0000-0002-9210-6544 Zhao Pengfei 1
Lv Han 1
Liu Wenhu 2
Wang Zhenchang 1 cjr.wzhch@vip.163.com
1 Department of Radiology, Beijing Friendship Hospital, Capital Medical University , Beijing , China
2 Department of Nephrology, Faculty of Kidney Diseases, Beijing Friendship Hospital, Capital Medical University , Beijing , China
Liu Feng
Electronic publication date: 2020 Apr 15
Publication date: 2020
Volume: 8
Electronic Location ID: e8989
Received 2019 Dec 30; Accepted 2020 Mar 26
Copyright: © 2020 Jin et al.
Copyright year: 2020
Copyright holder: Jin et al.
License: This is an open access article distributed under the terms of the Creative Commons Attribution License, which permits unrestricted use, distribution, reproduction and adaptation in any medium and for any purpose provided that it is properly attributed. For attribution, the original author(s), title, publication source (PeerJ) and either DOI or URL of the article must be cited.
License URL: https://creativecommons.org/licenses/by/4.0/

Keywords: End-stage renal disease, Hemodialysis, Arterial spin labeling (ASL), Amplitude of low-frequency fluctuation (ALFF), Neurovascular coupling, CBF/ALFF ratio, Across-voxel CBF-ALFF correlations, Amygdala, Cerebral blood flow (CBF), Neurovascular unit

Funding: National Natural Science Foundation of China 61527807, 81701644 and 61801311 Beijing Municipal Administration of Hospitals Mission Plan SML20150101 Beijing Scholars Program (2015) 160 Beijing Natural Science Foundation 7172064, 7162048 and 7182044 Beijing Municipal, Administration of Hospitals PX2018001 and QML20180103 Beijing Friendship Hospital, Capital Medical University YYZZ2017B01 This study was supported by the National Natural Science Foundation of China (61527807, 81701644 and 61801311), the Beijing Municipal Administration of Hospitals Mission Plan (SML20150101), the Beijing Scholars Program ((2015) 160), Beijing Natural Science Foundation (7172064, 7162048 and 7182044), the Beijing Municipal, Administration of Hospitals (PX2018001 and QML20180103), and the Beijing Friendship Hospital, Capital Medical University (YYZZ2017B01). The funders had no role in study design, data collection and analysis, decision to publish, or preparation of the manuscript.

==============================
Background

Altered cerebral blood flow (CBF) and amplitude of low-frequency fluctuation (ALFF) have been reported in hemodialysis patients. However, neurovascular coupling impairments, which provide a novel insight into the human brain, have not been reported in hemodialysis patients.

Methods

We combined arterial spin labeling (ASL) and blood oxygen level dependent (BOLD) techniques to investigate neurovascular coupling alterations and its relationships with demographic and clinical data in 46 hemodialysis patients and 47 healthy controls. To explore regional neuronal activity, ALFF was obtained from resting-state functional MRI. To measure cerebral vascular response, CBF was calculated from ASL. The across-voxel CBF–ALFF correlations for global neurovascular coupling and CBF/ALFF ratio for regional neurovascular coupling were compared between hemodialysis patients and healthy controls. Two-sample t-tests were used to compare the intergroup differences in CBF and ALFF. Multiple comparisons were corrected using a voxel-wise false discovery rate (FDR) method (P < 0.05).

Results

All hemodialysis patients and healthy controls showed significant across-voxel correlations between CBF and ALFF. Hemodialysis patients showed a significantly reduced global CBF–ALFF coupling (P = 0.0011) compared to healthy controls at the voxel-level. Of note, decreased CBF/ALFF ratio was exclusively located in the bilateral amygdala involved in emotional regulation and cognitive processing in hemodialysis patients. In hemodialysis patients, the decreased CBF (right olfactory cortex, anterior cingulate gyrus and bilateral insula) and ALFF (bilateral precuneus and superior frontal gyrus) were mainly located in the default mode network and salience network-related regions as well as increased CBF in the bilateral thalamus.

Conclusions

These novel findings reveal that disrupted neurovascular coupling may be a potential neural mechanism in hemodialysis patients.

Introduction

End-stage renal disease (ESRD) is the final stage of chronic kidney disease, and diagnosed as the estimated glomerular filtration rate <15 ml/min/1.7 m2 or the permanent functional point <10% of kidney capacity. ESRD is often accompanied by multiple-organ dysfunction, especially central nervous system and neurological symptoms (Brouns & De Deyn, 2004; Foley & Collins, 2007). There is high prevalence of cognitive dysfunction among hemodialysis patients compared with healthy controls (Bugnicourt et al., 2013; Kurella Tamura & Yaffe, 2011). Cognitive impairments (global cognition, attention, executive function and memory) may reduce the ability of patients in hemodialysis care, such as decreased medication adherence or lower life quality (Agganis et al., 2010; Cukor et al., 2009). Moreover, cognitive impairment is a common complication in ESRD patients with hemodialysis, leading to higher mortality and costly care (Agganis et al., 2010; Chen, Zhang & Lu, 2015). However, the mechanisms of neuro-pathophysiology of cognitive dysfunction in ESRD patients with hemodialysis remain largely unknown. Consequently, understanding neuropathological mechanisms of cognitive impairment in hemodialysis patients is crucial for improvement of diagnosis, treatment and follow-up.

Resting-state functional MRI is a useful tool for identifying functional abnormalities of cognitive complications in hemodialysis patients by analyzing blood oxygen level dependent (BOLD) signals. For example, amplitude of low-frequency fluctuation (ALFF) (Luo et al., 2016) and regional homogeneity (Chen et al., 2015) have been used to reveal spontaneous neuronal activity in ESRD patients. More recently, increasing evidence (Ma et al., 2016; Zhang et al., 2016) of disrupted resting-state brain network has been consistently detected in ESRD patients according to the methods of independent component analysis or nodal centrality (Li et al., 2016; Ling et al., 2014; Ma et al., 2015). The default mode network (DMN) is believed to be a well-established network in resting state, and typically includes the key nodes of the anterior cingulate and medial prefrontal cortex, posterior cingulate cortex and precuneus. Taken together, these core regions are involved in cognitive processing of the regulation between cognition and emotion, self-relevance, memory encoding, consolidation, motor activity and mind wandering (Buckner, Andrews-Hanna & Schacter, 2008; Greicius et al., 2003; Van den Heuvel & Hulshoff Pol, 2010). The salience network, comprised of dorsal anterior cingulate cortex and anterior insula, contributes to cognitive and behavioral control and participates in processing of directing attention (Liang et al., 2015; Shi et al., 2019). Furthermore, evidence from hemodialysis patients has shown altered amygdala-related functional connectivity (Chen et al., 2017; Li et al., 2018). The amygdala plays a critical role in interactive processing of affective and cognitive networks based on shared underlying neuro-circuitry (Mechelli et al., 2005; Soriano-Mas et al., 2013; Wu et al., 2017). Simultaneously, brain vascular supply can be assessed by cerebral blood flow (CBF) computed from arterial spin labeling (ASL) (Zheng et al., 2016). Using endogenous contrast, ASL is a non-invasive technique that can quantify CBF in a given time and explore the altered CBF in hemodialysis patients. Hemodialysis patients exhibited increased or decreased CBF in widespread regions of the brain, particularly decreased CBF in the prefrontal cortex (Cheng et al., 2019; Papoiu et al., 2014).

The human brain, which accounts for only 2% of body weight, consumes nearly 20% oxygen and glucose of the whole body, supporting spontaneous brain activity at resting-state (Phillips et al., 2016). As expected, there was a close relationship between CBF and brain metabolism, including glucose utilization, oxygen consumption, and aerobic glycolysis (Vaishnavi et al., 2010). Altered cerebral perfusion or neuronal activity in cerebral cortex was detected using only unimodal neuroimaging technique in several previous studies (Luo et al., 2016; Peng et al., 2018). However, these results could not fully reflect their close coupling situations. Thus, multimodal neuroimaging techniques were expected to detect cerebral coupling abnormalities in hemodialysis patients in this study. Several studies have investigated the close neurovascular coupling between CBF and neuronal activity in the normal brain (Phillips et al., 2016) and abnormal pathological conditions including Alzheimer’s disease (Tarantini et al., 2017), neuromyelitis optica (Guo et al., 2019), schizophrenia (Zhu et al., 2017) and type 2 diabetes mellitus (Hu et al., 2019).

In our study, BOLD data were used to measure ALFF value, and ASL perfusion data were explored to measure CBF value. Thus, we used ALFF to represent brain neuronal activity and CBF to reflect brain vascular response. Given the spatially different distribution of CBF and ALFF of the brain in hemodialysis patients and healthy controls, we hypothesized that hemodialysis patients could reveal a decreased global CBF–ALFF coupling compared with healthy subjects. Owing to the different affected regions in CBF and ALFF of the brain, we also hypothesized that hemodialysis patients could have specific affected regions of the CBF/ALFF ratio. Furthermore, we hypothesized that brain regions with increased/unchanged CBF and decreased ALFF could show an increased CBF/ALFF ratio, while brain regions with decreased/unchanged CBF and increased ALFF could show a decreased CBF/ALFF ratio in hemodialysis patients. To address these hypotheses, we collected ASL perfusion and resting-state BOLD data from 46 hemodialysis patients and 47 healthy controls. Global CBF–ALFF coupling analysis in gray matter and regional CBF/ALFF ratio analysis of each voxel were performed to compare the differences between hemodialysis patients and healthy controls. The results of this study are expected to provide an objective imaging biomarker and facilitate understanding of the neurovascular mechanism in hemodialysis patients from a new perspective.

Materials and Methods

Participants

The study protocol was approved by the Medical Research Ethics Committee of the Beijing Friendship Hospital (Approval Number: 2018-P2-158-02) and signed informed consent forms were obtained from all participants. A total of 111 right-handed participants (age ≥ 18 years) were enrolled in this study, including 58 patients with hemodialysis for at least 6 months (Liu et al., 2016; Yen et al., 2009) from the Department of Nephrology of Beijing Friendship Hospital and 53 healthy controls from the local community. The exclusion criteria were subjects with neuropsychiatric history, benign or malignant brain tumor history, cerebrovascular disease, a history of head trauma, other systemic diseases, renal transplant history, MRI contraindications, head motion, and poor imaging quality.

Demographic data including age and gender and hemodialysis times of each patient were retrieved and reviewed from the electronic medical records system. The measurements of brachial artery blood pressure (BP) of hemodialysis patients were performed at rest just before MRI scans and the average BP value of three consecutive measurements over ten minutes was recorded.

Laboratory examinations

Blood specimens of hemodialysis patients were drawn from the arterial end of the vascular access and were collected on the inter-dialytic (stable hemodialysis sessions) day before MRI scans to minimize the effect of potential bias. Serum levels of hemoglobin (g/l), creatinine (μmol/l), serum calcium and phosphorus (mmol/l), albumin (g/l), blood urea nitrogen (mmol/l) and serum ferritin (ng/ml) were measured and analyzed at the clinical laboratory of Beijing Friendship Hospital, with standard laboratory procedures using an automatic analyzer. The respective normal ranges of hemoglobin, creatinine, calcium, phosphorus, albumin, blood urea nitrogen and serum ferritin are 130–175 g/l, 41–111 μmol/l, 2.11–2.52 mmol/l, 0.85–1.51 mmol/l, 40.00–55.00 g/l, 2.60–7.50 mmol/l and 24.00–336.00 ng/ml in Beijing Friendship Hospital. No laboratory examination was performed in healthy controls.

Data acquisition

All imaging studies were performed at the Department of Radiology of Beijing Friendship Hospital. All MRI data were acquired using a 3.0-Tesla MR scanner (Discovery MR750, General Electric, Milwaukee, WI, USA) equipped with an eight-channel, phased-array head coil. Earplugs were used for noise reduction, and tight but comfortable sponge pads were used to minimize head motion. All subjects were asked to remain still, thoughtless and relaxed, keep their eyes closed, and stay awake during the scans.

Resting-state perfusion data were acquired by using a pseudo-continuous ASL sequence with background suppression. The parameters were as follows: repetition (TR) = 4852 ms; echo time (TE) = 10.7 ms; field of view (FOV) = 24 cm × 24 cm; matrix = 128 × 128; flip angle (FA) = 111°; slice thickness = 4.0 mm; no gap; 36 slices; post label delay = 2025 ms; number of excitation = 3. Resting-state BOLD data were acquired with the following parameters: TR = 2000 ms; TE = 35 ms; FOV = 24 cm × 24 cm; matrix = 64 × 64; FA = 90°; slice thickness = 5.0 mm; gap = 1 mm; 28 slices; 200 time-points. High-resolution three-dimensional T1-weighted images had the following parameters: TR = 8.492 ms; TE = 3.276 ms; inversion time (TI) = 450 ms; FOV = 24 cm × 24 cm; matrix = 256 × 256; FA = 15°; slice thickness = 1.0 mm; no gap; 196 slices.

CBF preprocessing

Using a single compartment model (Buxton et al., 1998), ASL difference images were computed after the deduction of the label images from the control images. The CBF maps were calculated from the combination with the ASL difference images and the proton density-weighted reference images (Xu et al., 2010). The detailed steps of CBF preprocessing were as follows (Liu, Zhuo & Yu, 2016). (1) Statistical Parametric Mapping (SPM8, https://www.fil.ion.ucl.ac.uk/spm/) software was used to normalize the CBF images to the standard Montreal Neurological Institute (MNI) space. (2) Individual ASL difference images of healthy controls were nonlinearly co-registered to the standard perfusion template in MNI space, and then averaged to generate a study-specific ASL template. (3) All individual ASL difference images of every subject were nonlinearly co-registered to this study-specific ASL template. (4) All CBF images were written into the standard space with the normalization parameters (derived from the previous step) and resliced into a 2 × 2 × 2 mm3 cubic voxel. (5) The co-registered CBF maps were removed of non-brain tissue and divided by the global mean CBF value of the gray matter. (6) The standardized images were spatially smoothed with a Gaussian kernel of 6 × 6 × 6 mm3 full-width at half maximum (FWHM).

Functional MRI data preprocessing

The functional MRI data preprocessing was performed in Data Processing Assistant for Resting-State Functional (DPARSF, http://rfmri.org/DPARSF) MRI software package. In total, 190 volumes (200 volumes were acquired, and the first 10 volumes were discarded for signal stabilization) were used for functional imaging data preprocessing. The following steps comprised slice timing, realignment, nuisance regressions, spatial normalization and smoothening (a 6 × 6 × 6 mm3 FWHM Gaussian kernel): (1) Slice time correction for the timing differences between slices and realignment for head-motion correction were performed. Head-motion thresholds were less than 3.0 mm of translation or 3.0° of rotation in any direction. Frame-wise displacement (FD) was also calculated to exclude the subjects with mean FD larger than 0.3 mm. No significant difference (P = 0.8138) was found in mean FD between the hemodialysis patients (0.0521 ± 0.0062) and healthy controls (0.0504 ± 0.0039). Then, mean FD was included as a covariate for the next statistical step. (2) The nuisance regressions (motion parameters based on the Friston-24 model, the average signals of the white matter and ventricular) were extracted as covariates. (3) In the normalization step, individual structural images were co-registered to the mean functional image; the transformed structural images were then segmented and spatially normalized to the MNI space using the Diffeomorphic Anatomical Registration Through the Exponentiated Lie algebra (DARTEL) technique (Ashburner, 2007).

For the ALFF analysis, the time course of each voxel was converted to the frequency domain using a Fast Fourier Transform. The averaged square root of the power spectrum (ALFF) value was then calculated across 0.01–0.08 Hz for all voxels in each participant. Finally, the ALFF value of each voxel was divided by the global brain mean ALFF value for standardization procedure (Liu et al., 2013).

Global CBF–ALFF coupling analysis

For each subject, Fisher’s z transformation was applied to both CBF and ALFF images for improving the normality of the data distribution and comparing across participants (Liu et al., 2017). The specific statistical processing was adopted from a previous study (Liang et al., 2013). For the global neurovascular coupling between CBF and ALFF, correlation analyses across voxels were performed for each subject. Because the neighboring voxels would be highly dependent due to spatial preprocessing such as registration and spatial smoothing, the effective degree of freedom (dfeff) in across-voxel correlation analyses would be much smaller than the number of voxels in the gray matter mask. Consequently, the dfeff in across-voxel correlation analyses was estimated using the following equation: dfeff=N(FWHMx×FWHMy×FWHMz)/ν−2

where ν is the volume of a voxel (2 × 2 × 2 mm3) and N is the number of voxels (N = 163,735) used in the analyses. FWHMx × FWHMy × FWHMz as the average smoothness of the CBF and ALFF images (9.7 × 10.7 × 10.2 mm3) was estimated using the software RESTplus (http://www.restfmri.net/). Therefore, the dfeff in across-voxel correlations was 1,235 in the present study. Consequently, a CBF–ALFF correlation coefficient value for each participant reflected the spatially consistent distribution between CBF and ALFF at gray matter level. Then, a two-sample t-test was used to compare the intergroup difference in CBF–ALFF correlation coefficients.

Regional CBF/ALFF ratio analysis

For the regional neurovascular coupling (blood supply per unit of neuronal activity), the CBF/ALFF ratio of each voxel was calculated for each subject. To identify significant difference in brain regions between the hemodialysis and healthy groups, a two-sample t-test was used to compare the differences in CBF/ALFF ratio, with age and gender as covariates. Multiple comparisons were performed by a voxel-wise false discovery rate (FDR) correction, with a corrected threshold of P < 0.05.

Voxel-wise comparisons in CBF and ALFF

To better understand the differences in CBF/ALFF ratio, voxel-wise comparisons were performed to detect the CBF and ALFF differences between the hemodialysis and healthy groups, by controlling age and gender as covariates. Multiple comparisons were also corrected using a voxel-wise FDR method (P < 0.05).

Clinical correlation analysis

Significant regions were extracted from regional CBF/ALFF ratio analysis and voxel-wise comparisons in CBF and ALFF between the two groups. Pearson’s correlation analyses were then performed to evaluate correlations between these regions and clinical variables (including hemodialysis times, blood pressure, hemoglobin, creatinine, calcium, phosphorus, albumin, blood urea nitrogen and serum ferritin) in the hemodialysis group. A P value < 0.05 was considered for significant differences.

Statistical analysis

Two-sample t-test was used to detect difference in age between the hemodialysis group and the control group. Chi-square test was used to demonstrate difference in gender between the two groups. All statistical analyses were performed using SPSS (an IBM Company, Chicago, IL, USA). Statistical significance was set as P < 0.05.

Results

Of the 111 participants (58 hemodialysis patients and 53 healthy controls), 18 participants were excluded due to the following reasons: one hemodialysis patient (left middle cerebral artery occlusion) and one healthy control (arteriovenous malformation) with cerebrovascular disease; one healthy control with benign brain tumor (cavernous hemangioma); six hemodialysis patients and one healthy control with head motion greater than 3.0 mm or 3.0° in any direction and three hemodialysis patients and three healthy controls with mean FD larger than 0.3 mm; two hemodialysis patients with claustrophobia. Finally, 46 hemodialysis patients and 47 healthy controls were included for analysis in this study.

Demographic data and clinical variables

The characteristic demographic data of hemodialysis patients and healthy controls were presented in Table 1. No significant differences were observed in age and gender between the two groups. The results of hemodialysis times, blood pressure and blood tests of the hemodialysis patients were also shown in Table 1.

Table 1 Demographic characteristics of HD and HC groups.

Variables	HD (n = 46)	HC (n = 47)	P value	
Age (years)	53.11 ± 1.58	55.57 ± 0.86	0.1263 (t = 1.543)a	
Gender (male/female)	28/18	22/25	0.1739 (χ2 = 1.849)b	
HD duration (HD times)	484.58 ± 186.31	–		
Blood pressure, mmHg		–		
Systolic pressure	145.91 ± 20.54	–		
Diastolic pressure	78.39 ± 13.72	–		
Blood tests		–		
Hemoglobin (g/l)	114.30 ± 12.15	–		
Creatinine (μmol/l)	935.01 ± 222.41	–		
Calcium (mmol/l)	2.35 ± 0.22	–		
Phosphorus (mmol/l)	1.46 ± 0.55	–		
Albumin (g/l)	43.55 ± 3.06	–		
Blood urea nitrogen (mmol/l)	20.55 ± 4.95	–		
Serum ferritin (ng/ml)	183.41 ±125.33	–		
Notes:

a Two-sample t-test.

b Chi-square test.

HD, hemodialysis patients; HC, healthy controls.

All values were expressed as mean ± standard deviation or number of subjects.

Global CBF–ALFF coupling changes in hemodialysis patients

Hemodialysis patients showed a significantly reduced global CBF–ALFF coupling (P = 0.0011) compared to healthy controls (Fig. 1). Significant difference was observed between two groups on CBF–ALFF correlation coefficients across voxels. In addition, all hemodialysis patients and all healthy controls showed significant across-voxel correlations between CBF and ALFF.

Figure 1 Reduced global CBF–ALFF coupling in hemodialysis patients.

Hemodialysis patients showed a significantly reduced (P = 0.0011) CBF–ALFF coupling compared to healthy controls. Significant difference was observed between two groups on CBF–ALFF correlation coefficients across voxels. Error bars represent the standard error. CBF, cerebral blood flow; ALFF, amplitude of low-frequency fluctuation; HC, healthy controls; HD, hemodialysis patients.

Spatial distribution of the CBF/ALFF ratio

The spatial distribution maps of CBF/ALFF ratio in healthy controls (Fig. 2A–2D) and hemodialysis patients (Fig. 2E–2H) were shown in Fig. 2. Although somewhat differences were found in brain regions and magnitudes, the two groups exhibited relatively consistent spatial distribution in these measures.

Figure 2 Spatial distribution maps of CBF/ALFF ratio in HC and HD.

Although somewhat differences were found in brain regions and magnitudes, HC group (A–D) and HD group (E–H) exhibited relatively consistent spatial distribution in these measures. (A) and (E), (B) and (F), (C) and (G) as well as (D) and (H) represent left lateral view, right lateral view, left medial view and right medial view, respectively. CBF, cerebral blood flow; ALFF, amplitude of low-frequency fluctuation; HC, healthy controls; HD, hemodialysis patients.

Regional CBF/ALFF ratio changes in hemodialysis patients

Compared to healthy controls, hemodialysis patients showed decreased CBF/ALFF ratio in left (Figs. 3A–3C) and right (Figs. 3D–3F) amygdala (P < 0.05, voxel-wise FDR correction). No significantly increased different regions were observed in regional CBF/ALFF ratio between two groups.

Figure 3 Group differences in CBF/ALFF ratio between hemodialysis patients and healthy controls.

Compared to healthy controls, hemodialysis patients showed decreased CBF/ALFF ratio in left (A–C) and right (D–F) amygdala (P < 0.05, voxel-wise FDR correction). (A) and (D), (B) and (E) as well as (C) and (F) represent coronal, axial and sagittal view, respectively. CBF, cerebral blood flow; ALFF, amplitude of low-frequency fluctuation; R, right; FDR, false discovery rate.

CBF and ALFF changes in hemodialysis patients

Compared with healthy controls, hemodialysis patients showed increased CBF in the thalamus bilaterally (Figs. 4A–4F) as well as decreased CBF in the right olfactory cortex and bilateral insula and anterior cingulated gyrus (Figs. 4G–4L) (P < 0.05, voxel-wise FDR correction) (Fig. 4; Table 2). Compared with healthy controls, hemodialysis patients exhibited increased ALFF in the amygdala bilaterally (Figs. 5A–5F) as well as decreased ALFF in superior frontal gyrus and precuneus bilaterally (Figs. 5G–5L) (P < 0.05, voxel-wise FDR correction) (Fig. 5; Table 3).

Figure 4 Group differences in CBF between hemodialysis patients and healthy controls.

Compared with healthy controls, hemodialysis patients showed increased CBF in the thalamus bilaterally (A–F) as well as decreased CBF in the right olfactory cortex and bilateral insula and anterior cingulated gyrus (G–L) (P < 0.05, voxel-wise FDR correction). CBF, cerebral blood flow; R, right; FDR, false discovery rate.

Table 2 Brain regions with significant group differences in CBF.

Regions	Cluster size (voxels)	Peak t values	Coordinates in MNI (x, y, z)	
HD < HC	
R olfactory cortex	226	−7.2060	18, 8, −22	
R insula	319	−7.8958	44, 14, −12	
L insula	219	−7.3120	−40, 16, −12	
R and L anterior cingulate gyrus	833	−7.4017	2, 36, 20	
HD > HC	
R thalamus	104	6.2191	18, −24, 2	
L thalamus	187	6.0539	−8, −18, 8	
Note:

Multiple comparisons were corrected by a voxel-wise FDR method with a corrected threshold of P < 0.05. CBF, cerebral blood flow; FDR, false discovery rate; MNI, Montreal Neurological Institute; R, right; L, left; HD, hemodialysis patients; HC, healthy controls.

Figure 5 Group differences in ALFF between hemodialysis patients and healthy controls.

Compared with healthy controls, hemodialysis patients exhibited increased ALFF in the amygdala bilaterally (A–F) as well as decreased ALFF in superior frontal gyrus and precuneus bilaterally (G–L) (P < 0.05, voxel-wise FDR correction). ALFF, amplitude of low-frequency fluctuation; R, right; FDR, false discovery rate.

Table 3 Brain regions with significant group differences in ALFF.

Regions	Cluster size (voxels)	Peak t values	Coordinates in MNI (x, y, z)	
HD < HC	
R and L superior frontal gyrus	308	−5.5620	−3, 51, 24	
R and L precuneus	281	−5.7403	9, −66, 51	
HD > HC	
R amygdala	169	5.6808	21, 3, −24	
L amygdala	175	4.8168	−24, 6, −21	
Note:

Multiple comparisons were corrected by a voxel-wise FDR method with a corrected threshold of P < 0.05. ALFF, amplitude of low-frequency fluctuation; FDR, false discovery rate; MNI, Montreal Neurological Institute; R, right; L, left; HD, hemodialysis patients; HC, healthy controls.

Correlation analysis of clinical variables

No significant difference was found in the correlations between each significant region and clinical variables (blood tests and hemodialysis times) in the hemodialysis and healthy groups (P > 0.05).

Discussion

To the best our knowledge, this is the first study to explore CBF–ALFF coupling alterations in hemodialysis patients by combining ASL and BOLD techniques. Compared with healthy controls, hemodialysis patients showed reduced across-voxel CBF–ALFF correlations of global neurovascular coupling. Moreover, hemodialysis patients revealed decreased CBF/ALFF ratio in the bilateral amygdala, which was involved in the emotion- and cognition-related regions. Furthermore, hemodialysis patients showed decreased CBF and ALFF bilaterally in DMN-related regions as well as increased CBF in the bilateral thalamus. These findings may provide objective neuroimaging evidence and improve the understanding of neural mechanisms from the perspective of neurovascular coupling in hemodialysis patients.

Consistent with our hypothesis, we found a significant across-voxel correlation between CBF and ALFF in healthy subjects, which was also reported in previous studies (Guo et al., 2019; Liang et al., 2013; Zhu et al., 2017), indicating the physiological neurovascular coupling in healthy brain. Additionally, across-voxel correlation between CBF and ALFF was also observed in hemodialysis patients, which was lower than that in healthy controls at the global level, presumably indicating global neurovascular decoupling in hemodialysis patients. Two possible explanations for global neurovascular decoupling in hemodialysis patients were as follows: (1) Chronic kidney disease has an impact on cerebrovascular hemodynamic alterations by changing CBF (Liu et al., 2018; Sedaghat et al., 2016). The changed CBF can lead to altered cerebral oxygen saturation, which is affected by hemodialysis duration, serum albumin concentration, and blood pH (Ito et al., 2015). Moreover, vessel impairment due to CBF alterations can lead to neurovascular decoupling. (2) Cerebral autoregulation, which is a complex control mechanism, maintains relatively constant CBF via changing cerebral vascular resistance in response to alterations of cerebral perfusion pressure or metabolic needs (Ghoshal & Freedman, 2019). The neurovascular coupling relies on the integrity of neurovascular unit, whose components are neurons, astrocytes and vasculature (Muoio, Persson & Sendeski, 2014). In this unit, astrocytes may play an important role in information exchange between neurons and vessels (Howarth, 2014; Stobart & Anderson, 2013), breaking the balance between neuronal activity and blood supply (Filosa et al., 2016). However, the impairment of neurovascular coupling in hemodialysis patient needs to be further explored.

Hemodialysis patients showed decreased CBF/ALFF ratio in the bilateral amygdala, which were the key nodes in emotional regulation and cognitive processing (Chen et al., 2017). In the present study, the bilateral amygdala showed normal CBF and increased ALFF, suggesting that the CBF/ALFF ratio reduction was mainly driven by the increased ALFF. This abnormality may at least partly account for the emotional and cognitive dysfunction in hemodialysis patients. The CBF/ALFF ratio maintains the balance between two aspects in the healthy brain (Guo et al., 2019). One aspect is vascular response and the other aspect is neuronal activity. The disruption in the CBF/ALFF ratio balance may lead to regional CBF/ALFF ratio enhancement or reduction. The increased CBF/ALFF ratio indicates redundant CBF per unit of neuronal activity or decreased ALFF with normal blood supply, whereas the decreased CBF/ALFF ratio indicates inadequate CBF per unit of neuronal activity or increased ALFF with normal blood supply. The deviation from the CBF/ALFF ratio balance in hemodialysis patients in this study is consistent with the latter situation. Moreover, the analysis of CBF/ALFF ratio can identify abnormal regions without significant intergroup differences in CBF or ALFF. Impairments of the amygdala in hemodialysis patients have been previously reported. One study (Li et al., 2018) combined structural covariance and functional connectivity analyses, and revealed amygdala-related pattern alterations of structural covariance and functional connectivity. The results demonstrated the reciprocal emotion–cognition interaction through amygdala-related emotional control network in hemodialysis patients. The authors also found that the hemoglobin level could directly lead to abnormal functional connectivity between amygdala and anterior cingulate cortex. Another study (Chen et al., 2017) showed a correlation between aberrant amygdala-based emotional regulatory circuits and depression in hemodialysis patients. Impaired neurocognitive performance was ascribed to the amygdala-related specific region in the hemodialysis patients. Nevertheless, the CBF/ALFF ratio is a new approach to measure regional alterations of neurovascular coupling in hemodialysis patients, and should be further investigated in the future.

Specifically, this study found significant intergroup differences in CBF and ALFF by voxel-wise analyses in several brain regions between hemodialysis patients and healthy controls. For example, hemodialysis patients exhibited decreased CBF in the right olfactory cortex and anterior cingulate gyrus, and decreased ALFF in the bilateral precuneus and superior frontal gyrus, most of which were components of the DMN. Several DMN-related regions showed decreased CBF in the inferior and middle frontal lobes, inferior temporal and parietal lobes (Cheng et al., 2019; Papoiu et al., 2014), decreased regional homogeneity in the bilateral frontal, parietal and temporal lobes (Chen et al., 2015), decreased gray matter volume in the bilateral precuneus and right frontal lobe (Zhang et al., 2013), reported by previous studies. Moreover, DMN-related decreases of functional connectivity have also been identified in hemodialysis patients, such as between medial prefrontal cortex and posterior cingulate cortex and precuneus (Ling et al., 2014). The DMN, which is a well-established network, has high metabolic activity at rest and is restrained during tasks (Buckner, Andrews-Hanna & Schacter, 2008). The DMN contributes to memory encoding, environmental monitoring, self-relevance and social functions, and regulates the interaction between cognitive functions and emotional processes (Greicius et al., 2003; Raichle & Snyder, 2007). Meanwhile, decreased CBF in the bilateral insular and anterior cingulate gyrus was identified in the hemodialysis group as compared to the healthy controls. Of note, the salience network typically comprises the bilateral insular and anterior cingulate gyrus. The insular, which is the key hub of the salience network, contributes to attentional, cognitive and behavioral functions (Menon & Uddin, 2010) and pain perception (Brooks & Tracey, 2007). The anterior cingulate gyrus, which is considered as a part of the cerebral limbic system, participates in emotional regulation (Bush, Luu & Posner, 2000), visual and auditory attention, language processing, memory and motor activity (Garrity et al., 2007). Furthermore, increased CBF was found in the bilateral thalamus of hemodialysis patients relative to healthy controls. The thalamus is regarded as a crucial relay station of multiple functional circuits regulating cognitive processing, including memory, emotion, attention and information processing (Fama & Sullivan, 2015). Taken together, DMN, salience network and thalamus were impaired in hemodialysis patients, which might explain the aberrant neurocognitive performance of these patients.

No significant difference was found in the correlations between each significant region and clinical variables (blood tests and hemodialysis times) in the hemodialysis and healthy groups. However, only hemoglobin levels were slightly below the normal range, while creatinine and blood urea nitrogen levels were substantially higher than the normal range due to the final stage of chronic kidney disease. Hemodialysis patients always suffer from anemia, malnutrition and protein-energy wasting conditions resulting from different pathogenic causes. Previous studies have shown that hemodialysis patients were usually vulnerable to under-nutrition caused by insufficient nutrient supply to the brain, resulting in brain abnormalities, such as cerebral hypoxia, decreased blood viscosity, brain hypo-perfusion and/or hypo-metabolism (Bornivelli et al., 2012; Yasuo et al., 2002). Therefore, long-term hemodialysis can significantly affect the cerebral circulation, function and metabolism, leading to poor mental health of hemodialysis patients due to low hemoglobin levels (Bornivelli et al., 2012). Other laboratory biochemical indicators exhibited a relatively normal range, and it was speculated that hemodialysis patients on regular use of vitamin D, calcitriol and/or phosphorus-chelating agents might have contributed to the steady state noted in the present study.

This study had several limitations. First, neuropsychological tests were not conducted to confirm different symptoms, so the relationship between the alterations of CBF/ALFF coupling and neurological complications could not be clarified, and the possibility of heterogeneity in hemodialysis patients could not be ruled out, which may affect the results. Second, CBF and ALFF indirectly reflect vascular response and neuronal activity, respectively. Thus, the analyses of the across-voxel CBF–ALFF correlations and CBF/ALFF ratio were indirect measurements of neurovascular coupling in hemodialysis patients. The technical limits may restrict the calculation of exact CBF and ALFF that result in alterations of neurovascular coupling in hemodialysis patients. Third, validation analyses were not performed in the present study. A larger sample size is required in further studies to validate the findings of the current study.

Conclusions

This study revealed neurovascular coupling impairments in hemodialysis patients via a combination of ASL and BOLD techniques. Specifically, decreased CBF/ALFF ratio in the bilateral amygdala was involved in emotional regulation and cognitive processing in hemodialysis patients. In addition, the DMN, salience network and thalamus were also impaired in hemodialysis patients, which might explain the aberrant neurocognitive performance of these patients. Consequently, these findings presented novel evidence that disrupted neurovascular coupling may be a potential neural mechanism in hemodialysis patients.

Supplemental Information

Supplemental Information 1 Raw data and software for nii format.

Data imagings (T map), subjects and clinical variables (CSV format) and software for nii format (mricron).

Click here for additional data file.

Additional Information and Declarations

Competing Interests

Author Contributions

Human Ethics

Data Availability

The authors declare that they have no competing interests.

Mei Jin conceived and designed the experiments, analyzed the data, prepared figures and/or tables, authored or reviewed drafts of the paper, and approved the final draft.

Liyan Wang conceived and designed the experiments, analyzed the data, prepared figures and/or tables, authored or reviewed drafts of the paper, and approved the final draft.

Hao Wang conceived and designed the experiments, analyzed the data, prepared figures and/or tables, authored or reviewed drafts of the paper, and approved the final draft.

Xue Han conceived and designed the experiments, analyzed the data, prepared figures and/or tables, authored or reviewed drafts of the paper, and approved the final draft.

Zongli Diao conceived and designed the experiments, prepared figures and/or tables, and approved the final draft.

Wang Guo conceived and designed the experiments, analyzed the data, prepared figures and/or tables, and approved the final draft.

Zhenghan Yang performed the experiments, prepared figures and/or tables, and approved the final draft.

Heyu Ding performed the experiments, prepared figures and/or tables, and approved the final draft.

Zheng Wang performed the experiments, analyzed the data, prepared figures and/or tables, and approved the final draft.

Peng Zhang performed the experiments, analyzed the data, prepared figures and/or tables, and approved the final draft.

Pengfei Zhao performed the experiments, prepared figures and/or tables, and approved the final draft.

Han Lv performed the experiments, analyzed the data, authored or reviewed drafts of the paper, and approved the final draft.

Wenhu Liu performed the experiments, authored or reviewed drafts of the paper, and approved the final draft.

Zhenchang Wang conceived and designed the experiments, authored or reviewed drafts of the paper, and approved the final draft.

The following information was supplied relating to ethical approvals (i.e., approving body and any reference numbers):

The present protocol was approved by the Medical Research Ethics Committee of the Beijing Friendship Hospital and signed informed consents were obtained from all subjects (2018-P2-158-02).

The following information was supplied regarding data availability:

The raw data are available in the Supplemental Files.

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
