# Peer review of "Disturbed neurovascular coupling in hemodialysis patients"

_PeerJ, doi:10.7717/peerj.8989_

## Round 0.1 · original submission · Major Revisions

The initial review indicates that the manuscript needs major revisions before it can be considered for publication in PeerJ.

·

Basic reporting

No comment.

Experimental design

No comment.

Validity of the findings

No comment.

Additional comments

Jin et al. aimed to examine the neurovascular coupling disturbance in hemodialysis patients by using a combination of ASL and resting-state fMRI approaches. The authors found that hemodialysis patients exhibited decreased global CBF-ALFF coupling as well as decreased CBF/ALFF ratio in the amygdala relative to healthy subjects, while no significant associations between imaging measures and clinical variables were observed. In general, the study is well performed and interesting, though more from a methodological aspect. The manuscript is clearly written and its presentation is logical. Even though the study may improve the understanding of brain function in hemodialysis, specific issues should be addressed in a potential revision.
1. The study lacks a clear hypothesis. The Introduction would benefit from giving an unequivocal hypothesis, e.g., increased or decreased coupling? the most affected regions?
2. The methodological description of blood tests should be more detailed like the MRI scanning.
3. Frame-wise displacement (FD) has a unit of mm.
4. If ALFF and fALFF are calculated, the preprocessing steps of fMRI data should not include band-pass filtered.
5. Fig. 1 should be revised to highlight the inter-group difference.
6. In Fig. 3, I suggest merging the bilateral amygdale into a brain map.
7. What do the colorbars represent in Fig. 3-5? T values?
8. Language needs attention or possibly an editor. For example, “To detect significant brain regions between two groups in CBF and ALFF value, multiple comparisons were performed with controlling age and gender as covariates…” Did the authors mean multiple comparisons were used to detect group differences?

Reviewer 2 ·

Basic reporting

no comment

Experimental design

no comment

Validity of the findings

no comment

Additional comments

This study has revealed the neurovascular coupling impairments in hemodialysis patients with a combination method between ASL and ALFF analysis, indicating a novel imaging phenotype of the hemodialysis patients; however, the diagnostic effect is insufficient and warrant future study. Notably, there are still some uncertain items which should be explained more clearly:
1. the comparison between hemodialysis patients and healthy controls was not enough to elucidate whether the metabolism of the amygdala was fluctuated by the kidney disease or dialysis. Actually, relevant studies have demonstrated the main disrupted brain activity pattern of the ESRD with dialysis was distributed in the core regions of the default mode network, and please give more reference as for this aberrance. Moreover, It is recommended to design one more group of the ERSD without dialysis  or analyze at different levels according to dialysis time.
2. The discussion is too tedious to propose the main findings. Better organised explanation of the discussion would be contributed to make a suitable prediction for the underlying neuropathological mechanism of hemodialysis. 
3.The six months of hemodialysis duration was regarded as for inclusion criteria, which need clarifying with some standard or reference.

Reviewer 3 ·

Basic reporting

See attached PDF

Experimental design

See attached PDF

Validity of the findings

See attached PDF

Additional comments

See attached PDF

Annotated reviews are not available for download in order to protect the identity of reviewers who chose to remain anonymous.

---

## Round 0.2 · Minor Revisions

Although all of three reviewers accepted the manuscript, there are still some minor concerns raised by the third reviewer in their attached PDF. The authors should carefully address these questions and check the whole manuscript.

·

Basic reporting

No comment.

Experimental design

No comment.

Validity of the findings

No comment.

Additional comments

All my concerns have been addressed. I am satisfied with the revision and this manuscript can be accepted.

Reviewer 2 ·

Basic reporting

no comment

Experimental design

no comment

Validity of the findings

no comment

Additional comments

The revised manuscript has been improved in experimental design and discussion. I have no comment about it and look forward to further study.

Reviewer 3 ·

Basic reporting

no comment

Experimental design

no comment

Validity of the findings

no comment

Annotated reviews are not available for download in order to protect the identity of reviewers who chose to remain anonymous.

---

## Round 0.3 · accepted · Accept

The authors have addressed all the concerns, and the manuscript can be accepted now.